# A combination of two human monoclonal antibodies cures symptomatic rabies

Guilherme Dias de Melo[1,†] , Florian Sonthonnax[1,2,†] , Gabriel Lepousez[3], Grégory Jouvion[4,5], Andrea Minola[6], Fabrizia Zatta[6], Florence Larrous[1], Lauriane Kergoat[1], Camille Mazo[3], Carine Moigneu[3], Roberta Aiello[7], Angela Salomoni[7], Elise Brisebard[4,8], Paola De Benedictis[7] , Davide Corti[6] & Hervé Bourhy[1,*] 

## Abstract

Rabies is a neglected disease caused by a neurotropic Lyssavirus, transmitted to humans predominantly by the bite of infected dogs. Rabies is preventable with vaccines or proper post-exposure prophylaxis (PEP), but it still causes about 60,000 deaths every year. No cure exists after the onset of clinical signs, and the case-fatality rate approaches 100% even with advanced supportive care. Here, we report that a combination of two potent neutralizing human monoclonal antibodies directed against the viral envelope glycoprotein cures symptomatic rabid mice. Treatment efficacy requires the concomitant administration of antibodies in the periphery and in the central nervous system through intracerebroventricular infusion. After such treatment, recovered mice presented good clinical condition, viral loads were undetectable, and the brain inflammatory profile was almost normal. Our findings provide the unprecedented proof of concept of an antibody-based therapeutic approach for symptomatic rabies.

**Keywords** immunotherapeutics; monoclonal antibody therapy; neglected diseases; neuroinfectious diseases; rabies virus

**Subject Categories** Immunology; Microbiology, Virology & Host Pathogen Interaction

## Introduction

Rabies is a lethal acute encephalomyelitis caused by a neurotropic Lyssavirus mainly transmitted to humans by the bite of domestic dogs (WHO, 2018). Rabies is an ancient illness (Tarantola, 2017), but it is still considered one of the most neglected diseases, especially in developing countries. The first vaccine against this infection was developed more than 130 years ago by Louis Pasteur (Bourhy *et al*, 2010), and today, rabies is fully preventable with proper post-exposure prophylaxis (PEP), with an estimation that 15–29 million patients exposed to rabies receive the PEP annually (WHO, 2018).

However, after the onset of clinical symptoms, which correlates with the presence of the virus in the central nervous system (CNS), rabies is nearly 100% fatal in infected patients, even with advanced supportive care (Dacheux *et al*, 2011; Jackson, 2018; Ugolini & Hemachudha, 2018; WHO, 2018). The mortality due to rabies is estimated to be about 60,000 deaths each year, mostly in Asia and Africa, and among them, 50% are children under 15 years of age (Hampson *et al*, 2015; WHO, 2018; Cantaert *et al*, 2019).

Several attempts have been performed to treat symptomatic rabies (Dacheux *et al*, 2011; Smith *et al*, 2019). In 2004, an infected patient from Wisconsin (USA) survived after a therapeutic approach that was named Milwaukee protocol (Willoughby *et al*, 2005). Since then, changes have been made in this protocol to arrive at its current version, which includes therapeutic coma, ketamine infusion, amantadine, and the management of cerebral vasospasm (Zeiler & Jackson, 2016). Nevertheless, its effectiveness is questionable since at least 31 documented failures have been reported (Zeiler & Jackson, 2016; Jackson, 2018).

In the quest for a novel therapeutic possibility, we have previously reported the selection of two human monoclonal antibodies (mAbs), RVC20 and RVC58, that were able to bind to two distinct antigenic sites on the RABV glycoprotein protein (sites I and III), to potently neutralize RABV isolates of all lineages, and of all phylogroup I non-RABV isolates, and that presented a protective

1  Lyssavirus Epidemiology and Neuropathology Unit, Institut Pasteur, Paris, France
2  Sorbonne-Paris Cité, Cellule Pasteur, Université Paris-Diderot, Paris, France
3  Perception and Memory Unit, Institut Pasteur, Paris, France
4  Experimental Neuropathology Unit, Institut Pasteur, Paris, France
5  INSERM, Pathophysiology of Pediatric Genetic Diseases, Sorbonne Université, Hôpital Armand-Trousseau, UF Génétique Moléculaire, Assistance Publique-Hôpitaux de Paris, Paris, France
6  Humabs BioMed SA, a subsidiary of Vir Biotechnology, Bellinzona, Switzerland
7  Istituto Zooprofilattico Sperimentale delle Venezie, Padua, Italy
8  Laboratoire d'Histopathologie, VetAgro-Sup, Université de Lyon, Lyon, France
   *Corresponding author. Tel: +33 1 45 68 87 85; E-mail: herve.bourhy@pasteur.fr
   †These authors contributed equally to this work

role when used as early PEP in hamsters as well (De Benedictis *et al*, 2016; Hellert *et al*, 2020).

Here, we show that a combination of the RVC20 and RVC58 monoclonal antibodies can effectively cure already symptomatic mice (late infection) when concomitantly administered both directly in the central nervous system, through intracerebroventricular infusion, and in the periphery, at the site of the infection. After such treatment, mice that survived the infection presented good clinical condition, the viral load was absent, and the inflammatory profile in the brain was close to that of uninfected animals. Altogether, our findings provide proof of concept that a targeted administration of human monoclonal antibodies represents a possibility with an unprecedented breadth and potency for the development of a low-risk product to treat rabies.

## Results and Discussion

### Peripheral immunotherapy is not efficient in curing rabies

To investigate whether the mAbs RVC20 and RVC58 display therapeutic activity against a lethal RABV infection *in vivo*, we set up a model of infection in Balb/c mice using a field RABV strain. In this model, the virus was detected from 4 days post-infection (dpi) in the spinal cord and at 5 dpi in the brain (Fig 1A). Weight loss and diminished motor performance are noticed from 7 dpi, while typical clinical signs (ruffled fur, lethargy, ataxia, paralysis) are detected from 8 dpi onwards (Fig 1B and C). Death occurred in all animals at 10–13 dpi (Fig 2A). In this condition, a single intramuscular (IM) injection of the 1:1 combination of RVC20/RVC58 (2 + 2 mg/kg) provided only modest survival effects when administered 2 or 4 dpi (Fig EV1), and no protective effect was observed when administered later. The blood half-life of the RVC20/RVC58 cocktail was determined as 6.16 days (Appendix Fig S1). At a higher dose (20 + 20 mg/kg), RVC20/RVC58 protected most animals from morbidity and mortality when administered 2 or 4 dpi, but displayed a limited success when treated 6 dpi (1/5). Intriguingly, some delayed deaths occurred in three treated animals (35, 55, and

68 dpi; Fig EV1B and C), which denoted the need to include late treatments to ensure the complete clearance of the remaining virus in the periphery, possibly avoiding a delayed wave of viral spread toward the CNS. Of note, we found similar results in CVS-11-infected golden Syrian hamsters treated just before the onset of clinical signs (Fig EV2). Altogether, these data support a dose-dependent effect of the mAbs cocktail and lead to the hypothesis that a strictly peripheral immunotherapy is not efficient in advanced rabies infection.

### Combined CNS and peripheral immunotherapy cures symptomatic rabies

We then further tested the therapeutic potential of RVC20/RVC58 on advanced phases of rabies virus infection, by combining IM injections with intracerebroventricular (ICV) administration in mice equipped with automated microinfusion pumps (Fig EV3A–C). The therapeutic protocol consisted of one IM injection of the RVC20/RVC58 cocktail (20 + 20 mg/kg) concomitantly with a continuous ICV infusion (2 + 2 mg/kg/day) during 20 days, starting at 6 dpi (presymptomatic phase, RABV already in the CNS; Fig 1A), at 7 dpi (prodromal phase, no clinical signs detected, but motor performance already impacted; Fig 1D and E), or at 8 dpi (symptomatic phase). At these time points, some cytokines and innate immune mediators' expression were already impacted (Appendix Fig S2 and Appendix Table S1). A second IM injection (20 + 20 mg/kg) two days after the end of the ICV infusion was administered to all treated animals.

IM + ICV treatment was 100% efficient in resolving the clinical signs and controlling the infection when started at 6 dpi (Figs 1 and 2). When started at 7 dpi, the treatment was able to promote survival and to ameliorate the clinical condition in 55.6% (5/9) of the infected animals, and when started at 8 dpi, the treatment was efficient in curing 33.3% (5/15) of the infected animals. Higher ICV doses of RVC20/RVC58 (10 + 10 mg/kg/day) did not increase survival rate (Figs 1 and 2). Remarkably, in the group of animals treated at 8 dpi, three animals which died during the ICV administration (16, 22, and 23 dpi) and another one that died after the end

---

**Figure 1.  Therapeutic efficacy of intramuscular and intracerebroventricular administration of the RVC20 and RVC58 monoclonal antibody cocktail in Tha-RABV-infected mice.**

A    Rabies virus infection kinetics in the central nervous system of infected mice. Tha-RABV load (copies/μg of RNA) detected in the thoracolumbar spinal cord, in the brainstem plus cerebellum, and in the cerebral cortex of infected mice according to different time points post-infection ($n = 4$ per time point). nd: not detected. Horizontal lines indicate the median.

B    Follow-up of the clinical signs of mice under different treatments. Heat maps were established based on a progressive 0–7 clinical score scale (0: no apparent changes; 1: ruffled fur; 2: slow movement, hind limb ataxia; 3: apathy; 4: monoplegia; 5: hind limb paralysis, tremors; 6: paralysis, conjunctivitis/keratitis, urine staining of the haircoat of the perineum; 7: death). Each line represents one animal throughout time.

C    Body weight progression of mice under different treatments. Mice were weighed on a daily basis during the treatment administration and then twice a week up to 100 days post-infection. All the mice were equipped with iPRECIO pumps weighing ca. 3.3 g. Each line represents one animal throughout time.

D, E  Mouse behavioral testing. The tested animals were non-infected ($n = 3$ mice with iPRECIO pump + $n = 4$ age-related mice without iPRECIO pump), infected non-treated ($n = 5$), infected and treated at 7 dpi ($n = 5$, mice #9 to #13), and infected and treated at 8 dpi ($n = 5$, mice #24 to #28). (D) Open-field test performed at 7 dpi (prodromal phase) and after the remission of clinical signs for the survivors (80 dpi). (E) Rotarod motor performance at 7 dpi (prodromal phase), 8 dpi (symptomatic phase), and after the remission of clinical signs for the survivors (20, 30, and 60 dpi). Horizontal lines indicate the median. Kruskal–Wallis followed by Dunn' multiple comparisons test (7 and 8 days post-infection) and Mann–Whitney test (20, 30, and 60 days post-infection). The rotarod performance presented a negative correlation (Spearman's $r = -0.768$) with the clinical score at 8 dpi. *$P < 0.05$, **$P < 0.01$, and ***$P < 0.001$. Treatment description = NI: non-infected ($n = 5$); NT: infected, non-treated ($n = 5$); infected, treated at 6 dpi (2 + 2 mg/kg) ($n = 4$); infected, treated at 7 dpi (2 + 2 mg/kg) ($n = 9$); infected, treated at 8 dpi (2 + 2 mg/kg) ($n = 15$); infected, treated at 8 dpi (10 + 10 mg/kg) ($n = 4$).

Exact *P* values are shown in Appendix Table S2.

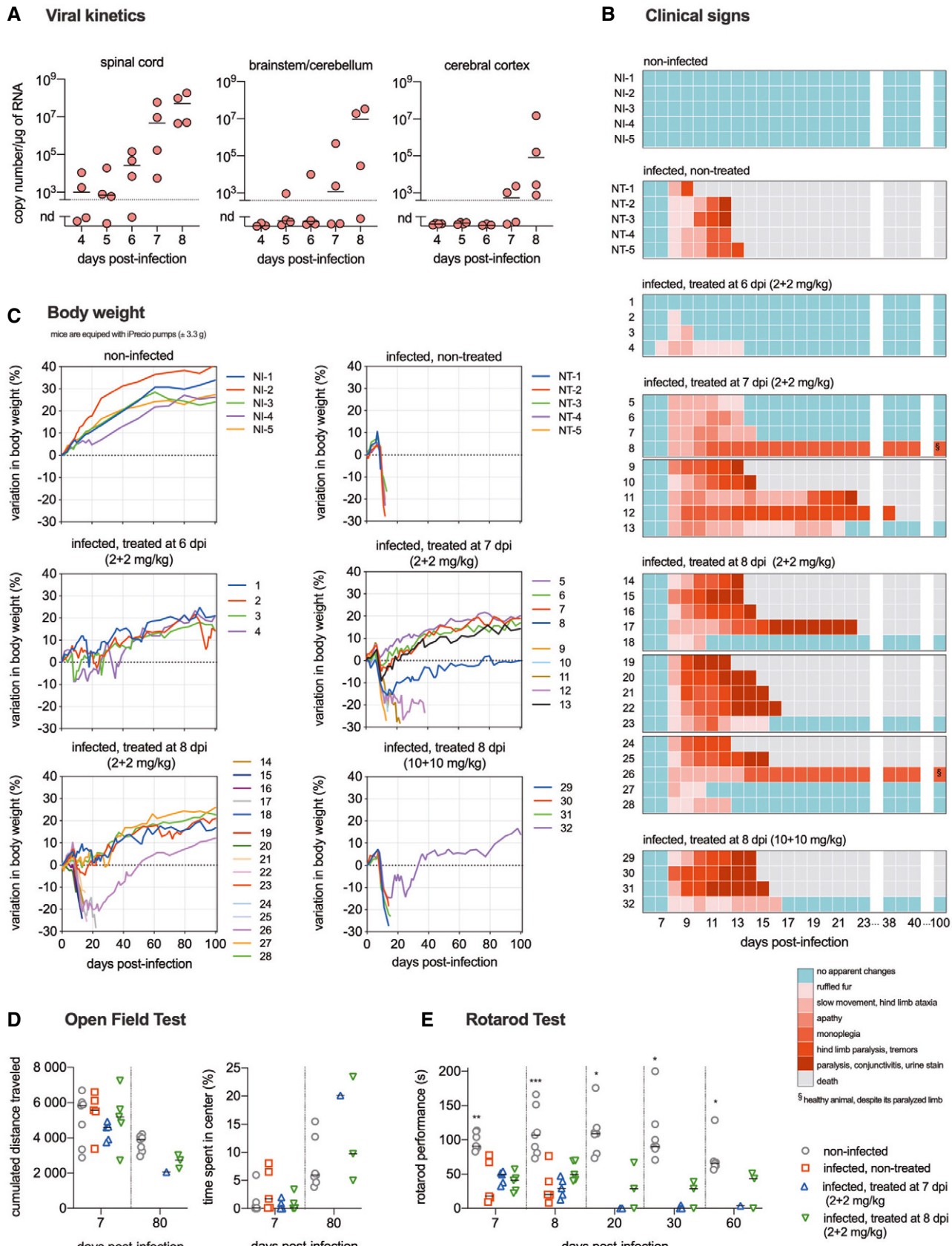

**Figure 1.**

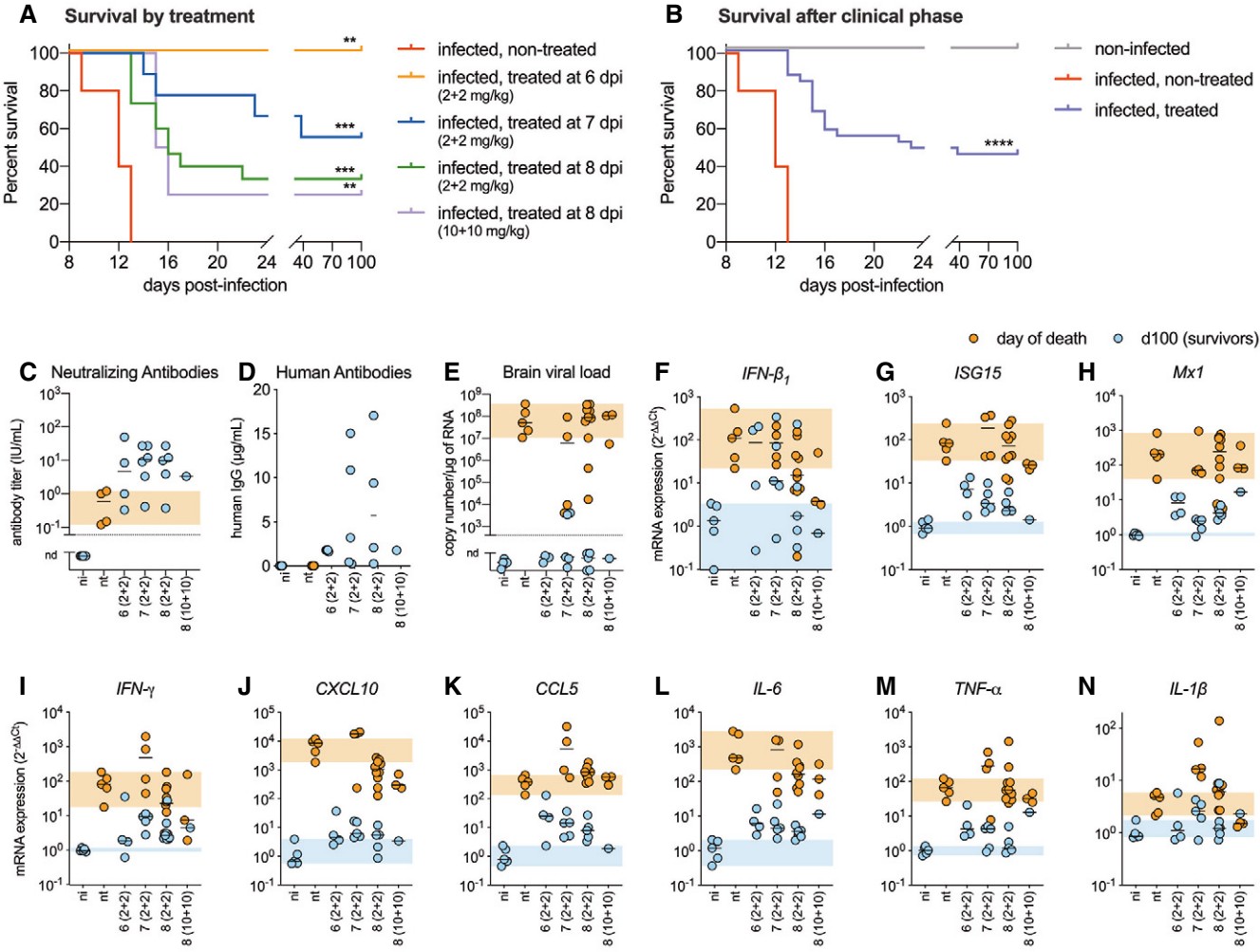

**Figure 2. Intramuscular and intracerebroventricular administration of the RVC20 and RVC58 monoclonal antibody cocktail cures symptomatic Tha-RABV-infected mice.**

A   Cumulative Kaplan–Meier survival curves of mice treated at different time points post-infection. Log-rank (Mantel–Cox) test to compare treated groups with the infected, non-treated group. **$P < 0.01$, ***$P < 0.001$. Treatment description = infected, non-treated ($n = 5$); infected, treated at 6 dpi (2 + 2 mg/kg) ($n = 4$); infected, treated at 7 dpi (2 + 2 mg/kg) ($n = 9$); infected, treated at 8 dpi (2 + 2 mg/kg) ($n = 15$); infected, treated at 8 dpi (10 + 10 mg/kg) ($n = 4$).

B   Cumulative Kaplan–Meier survival curves of all the treated animals that presented with clinical signs ($n = 31$, all treatments combined). Log-rank (Mantel–Cox) test to compare treated groups with the infected, non-treated group. ****$P < 0.0001$.

C   Virus-neutralizing antibodies detected in the serum of surviving mice from different experimental groups.

D   Residual human monoclonal antibodies in the serum of representative surviving mice at 100 dpi (LOD: limit of detection = 0.02 μg/ml).

E   Viral load in the brain of mice from different experimental groups. The samples were either collected at the time of death, or at 100 dpi for the survivors.

F–N  Cytokines and innate immune mediators' profile in the brain of mice from different experimental groups: (F) IFN-$\beta_1$, (G) ISG15, (H) Mx1, (I) IFN-$\gamma$, (J) CXCL10, (K) CCL5, (L) IL-6, (M) TNF-$\alpha$, and (N) IL-1$\beta$, detected in one brain hemisphere of mice from different experimental groups. The expression of the genes of interest was normalized to the GAPDH housekeeping gene.

Data information: Horizontal lines indicate the median. The orange crosshatched areas correspond to the 95% CI of the median from the infected and non-treated mice, and the blue crosshatched areas correspond to the 95% CI of the median from the non-infected mice. nd: not detected. Treatment description = ni: non-infected ($n = 5$); nt: infected, non-treated ($n = 5$); 6 (2 + 2): infected, treated at 6 dpi (2 + 2 mg/kg) ($n = 4$); 7 (2 + 2): infected, treated at 7 dpi (2 + 2 mg/kg) ($n = 9$); 8 (2 + 2): infected, treated at 8 dpi (2 + 2 mg/kg) ($n = 15$); 8 (10 + 10): infected, treated at 8 dpi (10 + 10 mg/kg) ($n = 4$). Exact $P$ values are shown in Appendix Table S2.

of the treatment (38 dpi) presented low viral load in their brains (Fig 2E), indicating that viral clearance by the RVC20/RVC58 cocktail had already started (Hunter *et al*, 2010). The *causa mortis* might be related to bad overall clinical condition associated with sequelae of the infection, such as brain damage or uncontrolled brain inflammation; intensive care in these cases could be suitable. Overall, combining all the treated animals, the IM + ICV treatment was

efficient in curing 45.2% (14/31) of mice that presented with clinical signs (Fig 2B).

Surviving animals were monitored until 100 dpi and did not develop further signs of disease, except two mice presenting permanent monoplegia despite an overall good condition and normal behavior (i.e., alert and active). Food intake and body weight became normal in the surviving mice after clinical phase (Fig 1C);

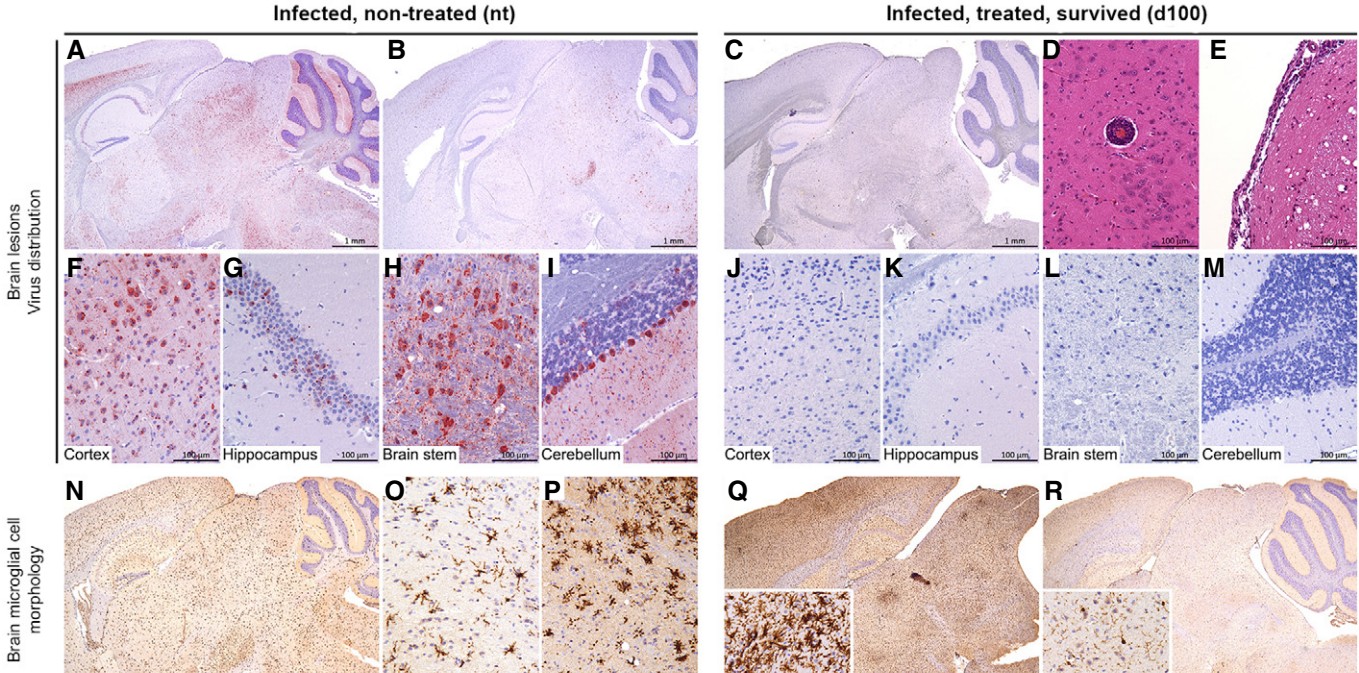

**Figure 3. Brain histopathological analysis in the brain of representative mice from different experimental groups.**

A–R  Histopathological analysis of the brain in Tha-RABV-infected non-treated mice (at 11 dpi) *versus* surviving mice (at 100 dpi) treated with intramuscular and intracerebroventricular administration of the RVC20 and RVC58 monoclonal antibody cocktail. The detection of Tha-RABV, using immunohistochemistry (IHC) on brain sections, revealed a heterogeneous invasion between the non-treated mice, ranging from extensive (A) to multifocal (B). Higher magnification of extensive invasion in the cortex (F), hippocampus (G), brain stem (H), and cerebellum (I). (N, O) Microglial cells were in general not reactive, but (P) small foci of reactive microglia (hyperplasia with thicker cell processes) could randomly be detected. (C, J, K, L, M) In contrast, even if a non-specific IHC signal was sometimes observed, no virus was detected on the brain sections of treated mice at 100 days post-infection. Only rare signs of neuro-inflammation were observed, as perivascular cuffing (D) (2/7 mice) and minimal meningitis (E) (3/7 mice). As for control non-treated mice, reactive microglial cells (Q) (2/7 mice) or resting microglia (R) were observed in treated mice. Representative images.

Data information: A–C and F–M: Tha-RABV IHC; N–R: Iba1 IHC to detect microglial cells; D–E: HE staining. F–I are high magnifications of A; J–M are high magnifications of C; and O–P are high magnifications of N.

spontaneous locomotor activity, exploratory behavior, and anxiety-related behavior, as monitored in the open-field test, were similar between controls and survivors (Fig 1D). Virus-neutralizing antibodies and human monoclonal antibodies were detected in the serum of surviving mice at 100 dpi (Fig 2C and D). No virus was detected in the brains of the surviving mice (except two animals with low but detectable viral load, Fig 2E), there was a long-lasting expression of antiviral mediators, and the inflammatory mediator profile in their brains was close to the one exhibited in the brain of non-infected mice (Figs 2F–N and EV4). Histological analysis of the brain of surviving mice revealed only rare signs of residual neuro-inflammation: perivascular cuffing (2/7), minimal meningitis (3/7), and reactive microglial cells (more marked in 2/7) (Fig 3A–R). Finally, we cannot attest that virus clearance occurred without neuronal loss, especially due to long-lasting deficit in the fine motor coordination of surviving mice, even if clinically healthy (Fig 1E).

**Fc portion plays role in rabies virus clearance**

To further understand whether the therapeutic activity of RVC20/RVC58 human mAbs relies also on Fc-dependent effector functions, we engineered the Fc of IgG1 RVC20/RVC58 to introduce the LALA mutation (L234A, L235A), that was shown to abrogate the binding to Fc-gamma receptor and complement, without compromising the *in vivo* pharmacokinetics (Stettler *et al*, 2016; Saunders, 2019). The RVC20-LALA/RVC58-LALA mAbs retained the same neutralizing activity as the non-mutated (wild-type) RVC20/RVC58 mAbs (Fig EV3D). However, the treatment with the RVC20-LALA/RVC58-LALA cocktail failed to protect mice when started at 8 dpi, and promoted only a 20% survival (1/5) when administered at 7 dpi (Fig EV3E–G, Appendix Fig S3), indicating that the Fc portion does play an important role in rabies virus clearance, as it seems to be the case in other viral infections (Chauhan *et al*, 2017), and in some observations of rabies virus clearance in mice (Hooper *et al*, 2009; Huang *et al*, 2014), dogs (Gnanadurai *et al*, 2013), and humans (de Souza & Madhusudana, 2014), where high titers of virus-neutralizing antibodies were detected in the cerebrospinal fluid. Analyzing the brain of the animals that succumbed to the infection during the course of the treatment, the ones that received the non-mutated RVC20/RVC58 mAbs tended to present lower viral loads and an intriguing pattern of pro-inflammatory cytokine expression, with increased levels of IL-1β and TNF-α, and reduced expression of IL-6 (Fig 2E–N). Conversely, in the brain of RVC20-LALA/RVC58-LALA mAbs-treated mice, viral loads and pro-inflammatory cytokine

expression were similar to those observed in non-treated animals (Appendix Fig S3). In a comparative manner, in neurodegenerative diseases, the antibody-mediated clearance of *tau* protein by microglia *in vitro* is dependent on Fc-gamma receptor binding (Andersson *et al*, 2019); Fc-gamma receptor is key to antibody uptake by neurons (Congdon *et al*, 2013). Consequently, the effective rabies virus clearance by the RVC20/RVC58 mAbs may be reinforced by Fc-gamma receptor binding, by permitting the access of the mAbs inside infected neurons, and by modulating microglial activity and inflammatory mediators' production, or even by promoting the infiltration of leukocytes (Quan *et al*, 2009; Chauhan *et al*, 2017; Huang & Sabatini, 2020).

The therapeutic activity of RVC20/RVC58 human mAbs against rabies relies on the innovative modality of administration, based on concomitant administrations on the site of infection (IM) and directly into the CNS (ICV), which block RABV spread and may render the clearance of the virus from the CNS possible, while modulating brain inflammation (Fig EV4). In addition, the infusion of mAbs unexpectedly stopped after 7 days of ICV deliverance in one mouse (#13); surprisingly, the animal survived, suggesting that the duration of ICV infusion could be shortened in some cases, naturally taking into consideration clinical status of the patient. ICV delivery is a safe administration route that distributes compounds throughout the brain, and despite the need of accurate neurosurgical techniques and to follow best practice requirements, it is extensively used in human neurology (Cohen-Pfeffer *et al*, 2017; Slavc *et al*, 2018; Duma *et al*, 2019). Notwithstanding, other routes to the CNS, such as intralumbar, should not be excluded and need to be assessed.

Rabies is fully preventable with proper post-exposure prophylaxis, and in this context, remarkable advances in the use of mAbs to replace human or equine rabies immunoglobulins (HRIG or ERIG) have been achieved, with one product already available in the Indian market (Rabishield®, a single mAb named as SII RMAb, 17C7, or RAB-1) (Gogtay *et al*, 2018). Another product has recently been authorized to be commercialized in India as well (Twinrab®, a cocktail of Docaravimab 62-71-3 and Miromavimab M777-16-3) (Kansagra *et al*, 2019), and other ones are currently under clinical trials (Sparrow *et al*, 2019). All these mAbs are directed against different antigenic sites of rabies glycoprotein. Despite these advances, none of them have been used as a therapeutic possibility to treat clinical rabies as RVC20/RVC58 cocktail. Consequently, we report the unprecedented discovery that neutralizing antibodies directed against the viral glycoprotein of rabies virus can effectively cure symptomatic rabid mice. These results provide proof of concept that this mAbs cocktail represents an opportunity to develop an effective treatment of CNS infection by rabies virus, especially if allied with more precise and earlier diagnosis of the infection.

# Materials and Methods

### Ethics statement

All mice experiments were performed in accordance with guidelines of the European and French guidelines (Directive 86/609/CEE and Decree 87–848 of 19 October 1987) and the Institut Pasteur Safety, Animal Care and Use Committee, and approved by the French Administration (Ministère de l'Enseignement et de la Recherche)

under the number O522-02. All hamster experiments were performed in strict accordance with the relevant national and local animal welfare bodies [Convention of the European Council No. 123 and National guidelines (Legislative Decrees 116/92 and 26/2014)]. The protocol was authorized by the Italian Ministry of Health (Decrees 128/2011-B and 115/2014-PR) before experiments were initiated and approved by the Committee on the Ethics of Animal Experiments of the IZSVe.

All animals were handled in strict accordance with good animal practice.

### Rabies model in Syrian hamsters and treatment

Female Syrian hamsters (*Mesocricetus auratus*) of 6–7 weeks of age (average weight 105 g) were purchased from Charles River Laboratories and handled in isolators under specific pathogen-free conditions, according to the institutional guidelines of the Central Animal Facility at IZSVe, with *ad libitum* access to water and food. Before any manipulation, animals underwent an acclimation period of 1 week.

Animals were challenged at day 0 by the intramuscular route (gastrocnemius muscle in the right hind limb) with 0.05 ml of an undiluted CVS-11 strain ($10^{6.76}$ MICLD$_{50}$/ml). Challenged animals were monitored after challenge twice a day for the observation of clinical signs, while body weight was noted once a day. The clinical signs were classified in a progressive 0–5 scale (0: no apparent changes; 1: right hind limb paresis; 2: bilateral hind limb paresis; 3: tremors and incoordination associated with mono- or bilateral paralysis. Animals with advanced paralysis were euthanized 1 day after the beginning of symptom if no regression was observed; 4: one of the previous scores associated with body weight loss. Animals displaying weight loss greater or equal to 20% were immediately euthanized; 5: death). Overall, clinical signs started 4 dpi and all animals were euthanized 5 dpi. At the time of the death (humane endpoints), animals were all euthanized by gas anesthesia (oxygen and 4% isoflurane) followed by $CO_2$ inhalation.

Thirty-six animals were divided into three groups of 12 animals each: (i) untreated, (ii) treated at 2 dpi, and (iii) treated at 3 dpi. A 1:1 combination of RVC20 and RVC58 mAbs (20 + 20 mg/kg) was administered in a final volume of 50 μl in the right hind limb. Six animals per group were observed up to 340 days, the remaining were sacrificed according to humane endpoints or 120 dpi, and the brain, the cerebellum and brainstem, and the spinal cord were collected separately for each animal and stored at −80°C. Viral mRNA was extracted using the kits NucleoSpin RNA Kit (Macherey-Nagel) and PolyATtract mRNA Isolation System III (Promega), and RT–qPCR was performed using the real-time platform CFX96 Real-Time PCR Detection System (Bio-Rad), as previously published (Hoffmann *et al*, 2010).

### Rabies model in mice

Eight-week-old female SPF Balb/cJRj mice were purchased from Janvier Laboratories and handled under specific pathogen-free conditions, according to the institutional guidelines of the Central Animal Facility at Institute Pasteur, with *ad libitum* access to water and food. Before any manipulation (surgery or infection), animals underwent an acclimation period of 1 week.

Animals were infected with 1,000 FFU (fluorescent focus units) of the pathogenic Tha-RABV (isolate 8743THA, EVAg collection, Ref-SKU: 014 V-02106, isolated from a human bitten by a dog in 1983 in Thailand) in a total volume of 100 µl, injected into the gastrocnemius muscle of both hind limbs (two injections of 25 µl in each limb). Thereafter, animals were monitored on a daily basis, with body weight and clinical signs noted. The clinical signs were classified in a progressive 0–7 scale (0: no apparent changes; 1: ruffled fur; 2: slow movement, hind limb ataxia; 3: apathy; 4: monoplegia; 5: hind limb paralysis, tremors; 6: paralysis, conjunctivitis/keratitis, urine staining of the haircoat of the perineum; 7: death). In the clinical phases of the disease, when the animals were paralyzed, DietGel Recovery (#72-06-5022, ClearH$_2$O) was used as diet supplement.

### Microinfusion pump activation and implantation to intracerebroventricular (ICV) delivery

iPRECIO microinfusion pumps (#SMP-300, Primetech Corp.) were connected to brain infusion cannula (Alzet Brain Infusion Kit 3, Durect Corporation) through a 9.0-cm tubing. The pump reservoirs were filled with filtered PBS and activated using the iPRECIO Management System (IMS-300) according to the manufacturer's instructions. The pumps were activated the day before the surgery using three infusion programs: (i) a flow rate of 0.1 µl/h, to keep the pump delivering PBS from the day of surgery on; (ii) a flow rate of 1.0 µl/h, 2 days before the treatment, to allow flow stability and dead volume elimination from the tubing (27.4 µl); and (iii) a flow rate of 1.0 µl/h during 20 days to deliver the treatment.

Mice were anesthetized intraperitoneally with 100 mg/kg ketamine (Imalgène 1000, Merial) and 10 mg/kg xylazine (Rompun, Bayer). The animals were shaved and placed in a stereotaxic frame coupled with a thermostat-regulated heating pad. To prevent eye dryness, an ophthalmic gel was applied on both eyes (Ocry-gel, TVM). Head was cleaned with iodine, 50 µl of 5 mg/kg lidocaine (Lurocaïne, Vetoquinol) was locally injected, a skin incision (about 1 cm) and removal of all soft tissue from the surface of the skull were performed, and a 1.0-mm hole was drilled through the skull with a battery-operated driller designed for rodent surgery (Geiger *et al*, 2008). The stereotaxic coordinates, taken bregma as reference, were −0.5 mm anteroposterior and +1.0 mm mediolateral. The 0.31 mm of diameter brain infusion cannula was inserted at −2.4 mm dorsoventral (from the skull surface) to deliver the treatment in the right lateral ventricle. The implants were cemented with cyanoacrylate ester (super glue gel + activator), and the pumps were implanted subcutaneously in the left dorsolateral site. The surgical wound was sutured with Prolene (#EH7975H, Ethicon), mice were injected subcutaneously with 0.1 mg/kg buprenorphine (Vetergesic, Ceva Santé Animale) in a final volume of 0.5 ml, and DietGel Recovery was offered. Following surgery, animals were individually housed with *ad libitum* access to food and water, and at least 1 week was allowed for recovery before virus challenge.

### Production of RVC20 and RVC58 antibodies by stably transfected CHO cells

The stably transfected RVC20- and RVC58-producing cell lines were developed using the GS Xceed Gene Expression System (Lonza). In particular, VH and VL sequences (synthetic genes codon-optimized for expression in hamster cells) were cloned into the pXC-IgG1 and pXC-Lambda vectors, respectively. The IgG1 allotype used is the G1m3 [corresponding to IMGT allele IGHG1*03; also known as G1m (f), which is the most frequent allele in the Caucasoid population], the lambda allotype is IGLC2*01, and the kappa allotype is IGKC*01 (k1m3). The single-gene vectors containing the VH and VL sequences of each antibody were then fused to form a double-gene vector that was subsequently linearized for the electroporation step (nucleofection using the Amaxa System Cell Line Nucleofector Kit V). The host cell line used here is CHOK1SV GS-KO (Lonza). Both alleles of the endogenous glutamine synthetase gene (GLUL, glutamate-ammonia ligase gene) have been "knocked out" by introduction of overlapping, out-of-frame deletions. The CHOK1SV GS-KO cell line is adapted for growth in suspension culture in a chemically defined, animal component-free medium. The generation of a stable cell line-producing RVC20 and RVC58 was based on the initial selection of high-producing uncloned cell lines that were grown in suspension and tested for productivity in an abridged fed-batch culture process. Supernatants were collected after 7-10 d, and IgG was affinity-purified by protein A chromatography (GE Healthcare) and desalted against PBS. The purified antibodies were quantified using Pierce Rapid Gold BCA Protein Assay Kit (Thermo Fisher Scientific).

RVC20 and RVC58 recombinant LALA variants (RVC20-LALA and RVC58-LALA) were produced as previously described (Stettler *et al*, 2016).

### PK profile of RVC20 and RVC58 monoclonal antibodies in mice

A mix of RVC20-rIgG1 (5 mg/kg) and RVC58-rIgG1 (5 mg/kg) was injected intravenously in 13-week-old B6/HuFcRn male (*n* = 3) and female (*n* = 1) mice. Blood samples were collected at 1, 4, 7, 13, and 18 days post-injection, the serum concentration of the RVC20/RVC58 antibody cocktail was measured, and the half-life was determined using GraphPad Prism.

### Rabies treatments in mice

Rabies treatment consisted of the concomitant intramuscular (IM) and ICV administration of the 1:1 combination of RVC20 and RVC58 human monoclonal antibodies. Six, seven, or eight days after RABV challenge, animals started receiving 2 + 2 mg/kg/day of RVC20 + RVC58 human mAbs diluted in Elliotts B solution (Lukare Medical) in a continuous flow rate of 1.0 µl/h by ICV route, during 20 days. The reservoirs were refilled with freshly diluted mAbs each 3–4 days. The ICV treatment was complemented by two IM administrations of mAbs in the site of the infection, one injection when the ICV deliverance of mAbs started (6, 7, or 8 dpi), and a second injection two days after the pumps stopped (28, 29, or 30 dpi). IM dose was 20 + 20 mg/kg in a final volume of 200 µl/mouse, administered into the gastrocnemius muscle of both hind limbs (two injections of 50 µl in each limb). Non-treated control mice received Elliotts B solution only.

### Open-field test

The open-field test is a test to measure spontaneous locomotor activity and exploratory activity and has also a component to test

fear/anxiety, as animals that avoid the central part of the field may be regarded as more anxious (Stanford, 2007). The animals were placed in the middle of the field (45 × 45 × 45 cm) at 7 dpi (prodromal phase) and after the remission of clinical signs for the surviving ones (80 dpi), and video-recorded during 5 min. The videos were then analyzed using the Icy software v.2.0.3.0 and the Single Mouse Tracker plugin (de Chaumont et al, 2012) using the "track white mice" option with a threshold of 120. The cumulated distance traveled was recorded, and in the same time, the floor zone was virtually divided into 16 squares (4 × 4) and the amount of time that each mouse passed in the four central squares was noted. The tested animals were non-infected ($n = 4$ mice without iPRECIO pump + $n = 3$ mice with iPRECIO pump), infected non-treated ($n = 5$), infected and treated at 7 dpi ($n = 5$, mice #9 to #13), and infected and treated at 8 dpi ($n = 5$, mice #24 to #28).

### Rotarod test

The rotarod is a test to measure motor performance (Jones & Roberts, 1968). It is constituted by a round rod that accelerates from 4 rpm to 40 rpm during 300 s. The time that animals keep moving on the rod is an indicator of motor capability. Mice were placed on an accelerating rotarod (Letica LE8200, Bioseb) at 7 dpi (prodromal phase), 8 dpi (symptomatic phase), and after the remission of clinical signs for the surviving ones (20, 30, and 60 dpi). The tested animals were non-infected ($n = 4$ mice without iPRECIO pump + $n = 3$ mice with iPRECIO pump), infected non-treated ($n = 5$), infected and treated at 7 dpi ($n = 5$, mice #9 to #13), and infected and treated at 8 dpi ($n = 5$, mice #24 to #28). Each animal was tested twice, with a 10-minute interval between each session, and the results are expressed as the mean latency time to fall off the rod. If the mouse was not able to move due to paralysis or apathy, their latency time was recorded as 0 s.

### Sampling

At the time of the death (humane endpoints) or at 100 days post-infection for the survivors, the animals were euthanized by $CO_2$ inhalation. Blood samples were collected, and the serum was recovered after coagulation and centrifugation at 2,000 g during 10 min at 4°C. Serum samples were stored at −20°C until analyses. Brains were removed and separated into two hemispheres: One hemisphere was fixed in 4% neutral-buffered formalin and the other hemisphere was stored at −80°.

### Rabies virus-neutralization assay

Rabies virus-neutralizing antibodies were detected using a modified rapid fluorescent focus inhibition test (RFFIT), recommended as the reference technique by WHO (Rupprecht et al, 2018). Briefly, mice sera were decomplemented at 56°C for 30 min and a constant dose of cell culture-adapted rabies challenge virus (CVS-11), determined by a previous titration to give a percentage of cell infection between 80% and 95%, was incubated with a threefold serial dilution of each serum. After incubation of the serum–virus mixtures for 1h at 37°C in a humid atmosphere under 5% $CO_2$, a suspension of BSR cells (a BHK-21 clone, kindly provided by Monique Lafon, Institut Pasteur, Paris) was added. After 24 h of incubation at 37°C in a

humid atmosphere under 5% $CO_2$, the cell monolayer was fixed with 80% (v/v) acetone and labeled with a fluoresceinated anti-rabies nucleocapsid antibody (5100, Merck). Rabies virus-neutralizing antibody titers in sera were calculated by comparison with a reference serum calibrated to the WHO reference serum. The threshold of detection was 0.06 IU/ml. An antibody titer above the threshold of 0.5 IU/ml was considered a proxy for protection (a so-called protective titer).

### RNA isolation and transcriptional analyses by quantitative PCR

Total RNA was extracted from the brains stored at −80°C using TRIzol and the phenol:chloroform method. Total RNAs were reverse-transcribed to first-strand cDNA using the QuantiTect Reverse Transcription Kit (#205311, Qiagen). qPCR was performed in a final volume of 20 μl per reaction in 96-well PCR plates using a thermocycler (7500t Real-time PCR system, Applied Biosystems). Briefly, 5 μl of cDNA (25 ng) was added to 15 μl of a master mix containing 10 μl of QuantiTect SYBR Green PCR Kit (#204143, Qiagen) and 5 μl of nuclease-free water with primers targeting the N gene at a final concentration of 0.5 μM (forward: 5′-CTG ACG TAG CAC TGG CAG AC-3′, reverse: 5′-AGT CGA CCT CCG TTC ATC AT-3′). The amplification conditions were as follows: 95°C for 15 min, 45 cycles of 94°C for 15 s and 60°C for 1 min, followed by a melt curve, from 60°C to 95°C. The gene of the nucleoprotein was used to quantification of the viral load in the brains, which was assessed by linear regression using a standard curve of eight known quantities of plasmids containing the nucleoprotein sequence (ranging from $10^8$ to $10^1$ copies). The threshold of detection was established as 400 viral copies/μg of RNA. The mouse gene targets were selected for quantifying host inflammatory mediators' transcripts in the brain using the GAPDH gene as a reference (#249900, Qiagen; QuantiTect Primer Assays IFN-$\beta_1$: QT00249662, ISG15: QT003 22749, Mx1: QT01064231, IFN-$\gamma$: QT02423428, CXCL10: QT0009 3436, CCL5: QT01747165, IL-6: QT00098875, TNF-$\alpha$: QT00104006, IL-1$\beta$: QT01048355, GAPDH: QT01658692). Variations in the gene expression were calculated as the n-fold change in expression in the brains from the infected mice compared with the brains of the uninfected ones using the $2^{-\Delta\Delta Ct}$ method (Pfaffl, 2001).

### Histological analysis

Brains were fixed in 10% neutral-buffered formalin and embedded in paraffin. Four-μm-thick sagittal sections (from olfactory lobes to cerebellum) were cut and stained to describe histological lesions (hematoxylin–eosin staining), or labeled to assess microglial cell morphology (anti-Iba1 IHC, 1:50, #01919741, Wako Chemical), or monitor Tha-RABV invasion (monoclonal rabbit anti-P49-1 antibody, 1:1,000; Sonthonnax et al, 2019). All IHC procedures were performed with the Bond III automat (Leica Biosystems). A blind histopathological analysis was carried out by two trained veterinary pathologists.

### Statistical analyses

Statistical analysis was performed with Prism v.8.2.1 (GraphPad). Differences between multiple groups were analyzed using Kruskal–Wallis followed by the Dunn's multiple comparisons test, and

**The paper explained**

**Problem**

Rabies is an almost invariably fatal disease. Despite being an ancient illness, rabies is still nowadays considered a neglected disease, being responsible of 60,000 estimated deaths each year, mainly related to young people coming from remote areas from developing countries. Rabies can be prevented by using vaccination and passive serotherapy, but currently no treatment is able to effectively cure rabies after the onset of the neurological symptoms.

**Results**

Here, we established a protocol to treat symptomatic rabies in mice using a cocktail of two potent neutralizing human monoclonal antibodies (mAbs RVC20/RVC58). The efficacy of this treatment is linked to the concomitant administration of these antibodies locally at the site of the infection and directly into the central nervous system. This therapy could lead to survival of infected mice, restoring their clinical condition and clearing rabies virus from their brain, with higher survival rates found in mice receiving the mAbs cocktail in early time points after the onset of the neurological symptoms.

**Impact**

Currently, despite some descriptions of patients surviving clinical rabies, there is no effective and reproducible therapy. Rabies can be prevented in rabies-exposed patients by the timely administration of a post-exposure prophylaxis (PEP) combining rabies vaccine and immunoglobulins. However, this PEP is not always accessible to target populations, and therefore, there is an urgent need for an affordable treatment against this infection to cure rabies-infected patients. The mAbs RVC20/RVC58 cocktail represents this unprecedented possibility to develop an effective treatment of brain infection by rabies virus, increasing survival rate and repairing neurological symptoms. Such a therapy would require a combined implementation of rapid and early diagnosis of rabies in infected patients as early administration of mAbs appears to be instrumental in the success of this therapeutic approach.

differences between two groups were assessed using Mann–Whitney test. Correlation between two variables was tested using Spearman's correlation test. Survival curves were compared using log-rank (Mantel–Cox) test. Multivariate statistical analysis (principal component analysis) to compare brain inflammatory mediators was performed using R v.3.6.2 (The R Foundation for Statistical Computing) and the FactoMineR package v.2.0 (Lê *et al*, 2008). If not indicated otherwise, data are expressed as the median and interquartile range. A *P*-value of < 0.05 was considered significant, and the exact *P* values are shown in Appendix Table S2. Due to the nature of the experiments (e.g., infusion pump programming, different treatment starts), they were performed unblinded and no randomization was used.

# Data availability

This study includes no data deposited in external repositories.

Expanded View for this article is available online.

## Acknowledgements

This research project received funds from the INFECT-ERA 2016 Project ANR 16-IFEC-0006-01 ToRRENT. This study was also partly funded by the Italian Ministry of Health (IZSVe RC 08/09) and by Vir Biotechnology. We thank David Hardy and Magali Tichit for their help with histology. We also thank Franco Mutinelli and Massimo Boldrin for ensuring animal welfare at IZSVe animal facilities, Elena Rota Nodari for animal handling, and Bianca Zecchin for molecular screening. Part of this work was performed at the UtechS Photonic BioImaging (PBI) platform, member of France Life Imaging network (grant ANR-11-INBS-0006).

## Author contributions

GDM, FS, GL, PDB, DC, and HB conceived and designed the experiments. GDM, FS, GL, FL, LK, CM, RA, AS, PDB, and HB performed the experiments. GDM, FS, LK, GJ, EB, and HB analyzed the data. GL, AM, FZ, PDB, DC, and HB contributed reagents/materials/analysis tools. GDM and HB wrote the manuscript.

## Conflict of interest

D.C., F.Z., and A.M. are employees of Vir Biotechnology Inc. and hold shares in Vir Biotechnology Inc.

## For more information

- Rabies section of the WHO website: https://www.who.int/health-topics/rabies
- Lyssavirus Epidemiology and Neuropathology unit at the Pasteur Institute: https://research.pasteur.fr/en/team/lyssavirus-epidemiology-and-neuropathology/

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
