## [Review Process File · EMBO Molecular Medicine]

A combination of two human monoclonal antibodies cures symptomatic rabies

Guilherme Melo, Florian Sonthonnax, Gabriel Lepousez, Grégory Jouvion, Andrea Minola, Fabrizia Zatta, Florence Larrous, Lauriane Kergoat, Camille Mazo, Carine Moigneu, Roberta Aiello, Angela Salomoni, Elise Brisebard, Paola De Benedictis, Davide Corti, and Hervé Bourhy

DOI: [10.15252/emmm.202012628](https://doi.org/10.15252/emmm.202012628)

Corresponding author: Hervé Bourhy (herve.bourhy@pasteur.fr)

Review Timeline:

Submission Date:	30th Apr 20
Editorial Decision:	22nd May 20
Revision Received:	10th Jul 20
Editorial Decision:	10th Aug 20
Revision Received:	26th Aug 20
Accepted:	27th Aug 20

Editor: Zeljko Durdevic

Transaction Report:

22nd May 2020

Dear Dr. Bourhy,

Thank you for the submission of your manuscript to EMBO Molecular Medicine. We have now heard back from the three referees who agreed to evaluate your manuscript. As you will see from the reports below, the referees acknowledge the interest of the study. However, they raise some concerns that should be addressed in a major revision of the present manuscript. Addressing the reviewers' concerns in full will be necessary for further considering the manuscript in our journal. Furthermore, I would like to suggest: 1) figure 2 is overcrowded, please split Figure 2 into 2 figures, e.g. Figure 3 could represent histological images, 2) please consider publishing your manuscript as a scientific report (3 Figures, ~22000 characters), for more information please check "Author Guidelines" <https://www.embopress.org/page/journal/17574684/authorguide#reportsarticleguide>.

Acceptance of the manuscript will entail a second round of review. Please note that EMBO Molecular Medicine encourages a single round of revision only and therefore, acceptance or rejection of the manuscript will depend on the completeness of your responses included in the next, final version of the manuscript. For this reason, and to save you from any frustrations in the end, I would strongly advise against returning an incomplete revision.

We realize that the current situation is exceptional on the account of the COVID-19/SARS-CoV-2 pandemic. Therefore, please let us know if you need more than three months to revise the manuscript.

I look forward to receiving your revised manuscript.

***** Reviewer's comments *****

Referee #1 (Remarks for Author):

Diaz de Mello et al report an antibody based therapeutic approach for symptomatic rabies. Using a mouse model of rabies infection, authors show that a combination of two potent neutralizing human monoclonal antibodies (mAbs) directed against the rabies viral envelope glycoprotein can

cure symptomatic rabid mice. The novelty of the work is the use of concomitant administration of anti-rabies mAbs in the periphery (intramuscular; i.m.) and in the central nervous system through intracerebroventricular infusion (ICV). These findings might have a potential medical impact as rabies is nearly 100% fatal in symptomatic infected patients. However, while results are significant, the survival rate of symptomatic infected mice remains moderate. The paper is clearly written. The work is scientifically sound. The methodology used is appropriate and the results support the conclusions drawn.

However, the manuscript could be further improved by addressing, or at least discussing, the following points:

1. i.m. +ICV mAb treatment (a combination of two mAbs RVC20/RVC58; 2+2 mg/kg/day) was 100% effective when started at 6 dpi. It promoted survival in 55.6% of the infected animals when started at 7 dpi, and in 33.3% of them when started at 8 dpi. Higher ICV doses of RVC20/RVC58 (10+10 mg/kg/day) did not increase survival rate, rather decreased it. This lack of protection might be related to higher viral- or inflammation induced brain damage at start of treatment. The authors showed brain inflammation of mice at day of death but not at early stages of infection. What are the differences in terms of brain damage (brain histopathological analysis) and inflammation (cytokines and innate immune mediators) at the different time points chosen to start mAb treatment (days 6, 7 and 8 p.i.)?

This is important to address as better characterization of inflammation/brain lesions might be key for therapeutic improvement.

2. Page 6. The authors report that some mice treated at day 8 p.i. eventually died despite a decrease in the viral load in their brains (Fig 2e), which reflected an effect of mAb treatment on viral clearance. They state that "intensive care in these cases could have been useful to protect mice from death". However, this remains speculative and other reasons could come into play. What about brain inflammation/damage in these mice?

3. Using the LALA mutant mAbs, the authors show that the Fc fragment plays an important role in the clearance of rabies virus. These results are important as they suggest that appropriate immune effectors might contribute to viral control. Such Fc-mediated viral control could involve several FcγR-expressing cells, either microglia cells/resident macrophages or recruited leukocytes (Huang and Sabatini, 2020, doi: 10.3389/fncel.2020.00065) . The authors might conduct mechanistic studies evaluating the role of microglia and/or innate immune cells in Fc-mediated viral control. It would also be important to show brain damage and inflammation in LALA-mutant mAb-treated mice. Such studies would strengthen the paper.

4. Page 6, authors state that "No virus was detected in the brains of the surviving mice", however 2 out of 4 surviving mice treated at 7 dpi, show low but detectable virus copy number in the brain.

5. Page 8. The authors mention that some mice received a reduced ICV deliverance of only seven days and discuss the protection of mouse 13 under this reduced treatment duration. However, they do not provide information on the total number of mice or the identification number of mice that received reduced ICV delivery. This needs to be clarified.

6. Figure 2 legend does not indicate when the histopathological analysis of the brains of untreated infected mice was performed. Overall, figure legends and supplemental figure legends are not sufficiently detailed (number of mice, time points, treatment of animals, ...). This should be improved.

7. The introduction and the discussion sections are very short and almost entirely focused on the RVC20/RVC58 anti-rabies mAbs developed by the authors. There is a first anti-rabies mAb that has recently gained regulatory approval in India (Rabishield) and several other candidates are being evaluated in clinical trials. The authors do not refer to those anti-rabies mAb. Nor do they discuss the differences/similarities between them and the RVC20/RVC58 mAbs in terms on breath and potency, antigenic site recognition, efficacy, administration,.... This information should be provided as it could be taken into consideration to better assess the medical impact of the authors' findings. It could also be useful for the non-specialized audience.

8. There are a few minor writing/editing mistakes (i.e. Supernatants "from" were collected, page 13).

Referee #2 (Remarks for Author):

Rabies is the most lethal of all infectious diseases. It has a typically long incubation period and there is no effective treatment once the virus begins replicating in the nervous system. Although rabies is generally well controlled in much of the world through vaccination of dogs and other animals, it still kills more than 50,000 people every year. Any effective therapy for symptomatic rabies would be a very important contribution.

This paper presents evidence that two monoclonal rabies neutralizing antibodies can be used as an effective cure for symptomatic rabies in mice. Intramuscular injection of the mAbs is not sufficient, but addition of intracranial mAb infusion generates a cure and clearance of the virus. The authors also present evidence that the protocol works in hamsters. If this protocol could be used to cure symptomatic rabies in humans, it would constitute a major medical advance. As the authors indicate, early detection of rabies infection will be critical for use of the protocol.

The paper is clear, concise, and the data are convincing.

Referee #3 (Remarks for Author):

To date, there is no effective clinical treatment for rabies once the symptoms appear. Melo et al here describe an interesting method to treat rabies by concomitant intramuscular (IM) administration of antibodies in the periphery and in the central nervous system through intracerebroventricular (ICV) infusion. Surprisingly, IM+ICV treatment was 100% efficient in resolving the clinical signs and controlling the infection when started at 6 dpi in a mouse model. Overall, this is a nice try for rabies treatment in an animal model, which sheds a light for the human rabies therapy. I am excited about this discovery, and it should be of great interest to the whole scientific community. Most experimental approaches are logical and solid, where necessary controls are included. But I have some concerns for this study:

Major ones:

1. When using a pair of human mono-Abs in a mouse model, is there any potential heterologous reaction, especially in the brain? What is the half-time for these human antibodies present in the mouse serum post IM inoculation? From Supplementary Fig. 1e-f, when used with a low dose (2+2

mg/kg) of these antibodies, the Ab titers in the serum began to decline at 6 dpi.

2. It is intriguing to apply ICV delivery strategy in rabies treatment. The advantage for this administration route is that the compounds can go throughout the brain. But looks like it is complicate to perform such a delicate surgery in a mouse. It is nice to present more pictures how to operate this surgery and what the mice looks like after the surgery. Another concern is that this surgery may change the BBB permeability and enhance neuroinflammation, which contributes to RABV clearance in the brain.

3. In my opinion, it is not necessary to conduct a continuous ICV infusion (2+2 mg/kg/day) during 20 days. Treatment with every two or three days might be enough. The authors have tried to reduce the duration from 20 to 7 days. However, only one mouse (mouse #13) is involved in this study. Thus, it is worth to repeat this experiment with more mice.

4. It is interesting to find that there are three delayed deaths occurred in very late time (35, 55 and 68 dpi). What is the Ab titer for each individual mouse at the day of death? The authors suggest that another IM injection with Abs at 2 days post ICV delivery is required to clear the remaining virus in the periphery. But, in my opinion, a shot of rabies vaccine may provide an even longer protection than Abs, considering the relatively short half-time of Abs in the mouse serum.

5. The authors revealed that the Fc part of Abs were required for the clearance of RABV in the brain by introduce the LALA mutation that was shown to abrogate the binding to Fc-gamma receptor. But the readers still don't know what the detailed mechanism is. At least, they should extend these observations in the discussion part.

6. In Supplemental Fig. 4, the authors performed component analysis of immune mediators in the infected-brains and showed the correlation among the nine immune mediators in mouse brains. However, in the methods and figure legend, we can't find any detailed information about this analysis. Thus, it is really difficult to figure out what is mean about PCA, PC1 or PC2. Also, is there any references showing that ISG15 and Mx1 play a role in restricting RABV infection?

Minor ones:

1. The line numbers are missing in the manuscript, so it is difficult to pinpoint the questions in the text.

2. Some descriptions in the figure legends including the mouse numbers, infection dose etc. are not clear enough and the readers need to come back to the results/methods from time to time.

3. In Fig. 1c, the percentage of body weight (present body weight/original body weight)is easy to read.

4. In Fig. 1d&e, some of the statistics are missing.

5. In Supplementary Fig. 1c and Fig. 3c, the statistics are missing.

Referee #1 (Remarks for Author):

Diaz de Mello et al report an antibody based therapeutic approach for symptomatic rabies. Using a mouse model of rabies infection, authors show that a combination of two potent neutralizing human monoclonal antibodies (mAbs) directed against the rabies viral envelope glycoprotein can cure symptomatic rabid mice. The novelty of the work is the use of concomitant administration of anti-rabies mAbs in the periphery (intramuscular; i.m.) and in the central nervous system through intracerebroventricular infusion (ICV). These findings might have a potential medical impact as rabies is nearly 100% fatal in symptomatic infected patients. However, while results are significant, the survival rate of symptomatic infected mice remains moderate. The paper is clearly written. The work is scientifically sound. The methodology used is appropriate and the results support the conclusions drawn.

However, the manuscript could be further improved by addressing, or at least discussing, the following points:

1. i.m. +ICV mAb treatment (a combination of two mAbs RVC20/RVC58; 2+2 mg/kg/day) was 100% effective when started at 6 dpi. It promoted survival in 55.6% of the infected animals when started at 7 dpi, and in 33.3% of them when started at 8 dpi. Higher ICV doses of RVC20/RVC58 (10+10 mg/kg/day) did not increase survival rate, rather decreased it. This lack of protection might be related to higher viral- or inflammation induced brain damage at start of treatment. The authors showed brain inflammation of mice at day of death but not at early stages of infection. What are the differences in terms of brain damage (brain histopathological analysis) and inflammation (cytokines and innate immune mediators) at the different time points chosen to start mAb treatment (days 6, 7 and 8 p.i.)?

This is important to address as better characterization of inflammation/brain lesions might be key for therapeutic improvement.

The profile of cytokines and innate immune mediators' gene expression are already impacted by 6 dpi, notably IFN-beta1, ISG15, CCL5 and TNF-alpha, while other mediators presented differences in their gene expression in the brain according to the time post-infection. By histopathology and immunohistochemistry, on the other hand, remarkable changes were noticed at 8 dpi, but not before. We added this info in lines 135-137 and in Appendix Fig S2 and Appendix Table S1.

2. Page 6. The authors report that some mice treated at day 8 p.i. eventually died despite a decrease in the viral load in their brains (Fig 2e), which reflected an effect of mAb treatment on viral clearance. They state that "intensive care in these cases could have been useful to protect mice from death". However, this remains speculative and other reasons could come into play. What about brain inflammation/damage in these mice?

We addressed this issue in lines 148-151

3. Using the LALA mutant mAbs, the authors show that the Fc fragment plays an important role in the clearance of rabies virus. These results are important as they suggest that appropriate immune effectors might contribute to viral control. Such Fc-mediated viral control could involve several FcγR-expressing cells, either microglia cells/resident macrophages or recruited leukocytes (Huang and Sabatini, 2020, doi:10.3389/fncel.2020.00065). The authors might conduct mechanistic studies evaluating the role of microglia and/or innate immune cells in Fc-mediated viral control. It would also be important to show brain damage and inflammation in LALA-mutant mAb-treated mice. Such studies would strengthen the paper.

We discussed this data in lines 183-191. Data regarding brain inflammation in LALA mAbs-treated animals was added in Appendix Fig S3.

4. Page 6, authors state that "No virus was detected in the brains of the surviving mice", however 2 out of 4 surviving mice treated at 7 dpi, show low but detectable virus copy number in the brain.

We completed this info in lines 160-161.

5. Page 8. The authors mention that some mice received a reduced ICV deliverance of only seven days ~~discuss~~ the protection of mouse 13 under this reduced treatment duration. However, they do not provide information on the total number of mice or the identification number of mice that received reduced ICV ~~delivery~~. This needs to be clarified.

The animal #13 was the only animal that received a 7-day ICV treatment. Actually, we observed that the pump stopped the mAbs deliverance after 7 days without a known cause (problems with the programming, battery, plug inside the tube).

We rewrote this phrase (lines 197-200) to clarify this information and to avoid any misunderstandings.

6. Figure 2 legend does not indicate when the histopathological analysis of the brains of untreated infected ~~was~~ performed. Overall, figure legends and supplemental figure legends are not sufficiently detailed (number of mice, time points, treatment of animals, ...). This should be improved.

We completed the information in the figure legends.

7. The introduction and the discussion sections are very short and almost entirely focused on the RVC20/RVC58 anti-rabies mAbs developed by the authors. There is a first anti-rabies mAb that has recently gained regulatory approval in India (Rabishield) and several other candidates are being evaluated in clinical trials. The authors ~~refer~~ to those anti-rabies mAb. Nor do they discuss the differences/similarities between them and the RVC20/RVC58 mAbs in terms on breath and potency, antigenic site recognition, efficacy, administration,... This information should be provided as it could be taken into consideration to better assess the medical impact of the authors' findings. It could also be useful for the non-specialized audience.

We addressed this issue in the discussion section (lines 207-217).

8. There are a few minor writing/editing mistakes (i.e. Supernatants "from" were collected, page 13).

We corrected this phrase.

Referee #2 (Remarks for Author):

Rabies is the most lethal of all infectious diseases. It has a typically long incubation period and there is no effective treatment once the virus begins replicating in the nervous system. Although rabies is generally well controlled in much of the world through vaccination of dogs and other animals, it still kills more than 50,000 people every year. Any effective therapy for symptomatic rabies would be a very important contribution.

This paper presents evidence that two monoclonal rabies neutralizing antibodies can be used as an effective cure for symptomatic rabies in mice. Intramuscular injection of the mAbs is not sufficient, but addition of intracranial mAb infusion generates a cure and clearance of the virus. The authors also present evidence that the protocol works in hamsters. If this protocol could be used to cure symptomatic rabies in humans, it would constitute a major medical advance. As the authors indicate, early detection of rabies infection will be critical for use of the protocol.

The paper is clear, concise, and the data are convincing.

We thank the referee for this review.

Referee #3 (Remarks for Author):

To date, there is no effective clinical treatment for rabies once the symptoms appear. Melo et al here describe an interesting method to treat rabies by concomitant intramuscular (IM) administration of antibodies in the periphery and in the central nervous system through intracerebroventricular (ICV) infusion. Surprisingly, IM+ICV treatment was 100% efficient in resolving the clinical signs and controlling the infection when started at 6 dpi in a mouse model. Overall, this is a nice try for rabies treatment in an animal model, which sheds a light for the human rabies therapy. I am excited about this discovery, and it should be of great interest to the whole scientific community. Most experimental approaches are logical and solid, where necessary controls are included. But I have some concerns for this study:

Major ones:

1. When using a pair of human mono-Abs in a mouse model, is there any potential heterologous reaction, especially in the brain? What is the half-time for these human antibodies present in the mouse serum post IM inoculation? From Supplementary Fig. 1e-f, when used with a low dose (2+2 mg/kg) of these antibodies, the Ab titers in the serum began to decline at 6 dpi.

There will always be a potential risk of heterologous reaction when using human antibodies in mice. However, we have included a group of non-infected mice that were treated with the mAbs cocktail and no changes were noticed. Specifically, the binding strengths of human IgG to mouse Fc-gamma receptors are similar to those related to human Fc-gamma receptors (Dekkers et al., 2017). Further, as comparative, a study concerning toxicity of a humanized mAb administered intrathecally in cynomolgus monkeys showed no side effects of mAbs injection in the CSF (Braen et al., 2010).

Regarding the levels of mAbs in the mouse serum indeed, the titers were not so high in mice treated with 1 IM inoculation (Figure EV1F). This is why we decided to perform a second IM injection in the ICV treated animals. In this case, even with lower levels, there are reminiscent mAbs at 100 dpi in the serum of surviving mice after ICV treatment (Figure 2D). The blood half-life of the RVC20/RVC58 cocktail is about 6.16 days; we added this info in lines 114-115, lines 343-348 and in Appendix Fig S1.

Braen APJM, Perron J, Tellier P, Catala AR, Kolaitis G, Geng W (2010) A 4-Week Intrathecal Toxicity and Pharmacokinetic Study With Trastuzumab in Cynomolgus Monkeys. *International Journal of Toxicology* 29: 259-267

Dekkers G, Bentlage AEH, Stegmann TC, Howie HL, Lissenberg-Thunnissen S, Zimring J, Rispen T, Vidarsson G (2017) Affinity of human IgG subclasses to mouse Fc gamma receptors. *mAbs* 9: 767-773

2. It is intriguing to apply ICV delivery strategy in rabies treatment. The advantage for this administration route is that the compounds can go throughout the brain. But looks like it is complicated to perform such a delicate surgery in a mouse. It is nice to present more pictures how to operate this surgery and what the mice look like after the surgery. Another concern is that this surgery may change the BBB permeability and enhance neuroinflammation, which contributes to RABV clearance in the brain.

We used a standard stereotaxic setup with predefined coordinates to achieve the lateral ventricle (description in lines 298-316). Besides the picture in Figure EV3A, we can attach other images here below for your information, but we would prefer not to include them in the manuscript.

The infection of mice was performed no earlier than 7 days after the surgical implantation of the ICV pumps to allow an appropriate wound healing. Further, infected/non-treated and non-infected/non-treated mice underwent the same surgical procedures. Of note, in human patients, a minimum of 5-days recovery between ICV devices implantation and treatment beginning is preconized (Cohen-Pfeffer et al. 2017).

Cohen-Pfeffer JL, Gururangan S, Lester T, Lim DA, Shaywitz AJ, Westphal M, Slavic I (2017) Intracerebroventricular Delivery as a Safe, Long-Term Route of Drug Administration. *Pediatric Neurology* 67: 23-35

3. In my opinion, it is not necessary to conduct a continuous ICV infusion (2+2 mg/kg/day) during 20 days. Treatment with every two or three days might be enough. The authors have tried to reduce the duration from 20 to 7 days. However, only one mouse (mouse #13) is involved in this study. Thus, it is worth to repeat this experiment with more mice.

Actually, according to Figure 1B, several treated animals died around 14-15 dpi, which corresponds to 7 days of treatment. This does not encourage an experiment with only 7 days. Further, due to the rapid cerebrospinal fluid turnover in mice of approximately 2 hours (Calias et al. 2014) we did not consider the use of intermittent mAbs ICV infusion.

Regarding animal #13, we observed that its iPrecio pump unexpectedly stopped the mAbs deliverance after 7 days; we do not know the reason of this interruption (possibilities: problems with the programming, battery, plug inside the tube).

We rewrote this phrase (lines 197-200) to avoid any misunderstandings.

Calias P, Banks WA, Begley D, Scarpa M, Dickson P (2014) Intrathecal delivery of protein therapeutics to the brain: A critical reassessment. *Pharmacology & Therapeutics* 144: 114-122

4. It is interesting to find that there are three delayed deaths occurred in very late time (35, 55 and 68 dpi). What is the Ab titer for each individual mouse at the day of death? The authors suggest that another IM injection with Abs at 2 days post ICV delivery is required to clear the remaining virus in the periphery. But, in my opinion, a shot of rabies vaccine may provide an even longer protection than Abs, considering the relatively short half-time of Abs in the mouse serum.

These 3 delayed deaths were detected in the animals treated by one IM injection (Figure EV1). Their neutralizing antibody titer and their human antibodies levels (mAbs) is similar to the other animals in the same groups (Figure EV1-EF).

The use of rabies vaccine to increase the protection is indeed an interesting idea and should be taken in consideration to refine the therapeutic protocol, nevertheless, in these experiments we would like to test the efficacy of the mAbs cocktail alone in treating the infected mice.

5. The authors revealed that the Fc part of Abs were required for the clearance of RABV in the brain by introduce the LALA mutation that was shown to abrogate the binding to Fc-gamma receptor. But the readers

still don't know what the detailed mechanism is. At least, they should extend these observations in the discussion part.

We discussed this data in lines 183-191.

6. In Supplemental Fig. 4, the authors performed component analysis of immune mediators in the infected-brains and showed the correlation among the nine immune mediators in mouse brains. However, in the methods and figure legend, we can't find any detailed information about this analysis. Thus, it is really difficult to figure out what is mean about PCA, PC1 or PC2. Also, is there any references showing that ISG15 and Mx1 play a role in restricting RABV infection?

This info is already present in the methods section, under the Multivariate statistical analysis part (lines 455-437). We completed the legend of Fig EV4.

Regarding ISG15 (Zhao et al. 2017) and Mx1 (Leroy et al. 2006), both mediators have been described to play a role in rabies virus replication.

Leroy M, Pire G, Baise E, Desmecht D (2006) Expression of the interferon-alpha/beta-inducible bovine Mx1 dynamin interferes with replication of rabies virus. *Neurobiology of Disease* 21: 515-521
Zhao P, Jiang T, Zhong Z, Zhao L, Yang S, Xia X (2017) Inhibition of rabies virus replication by interferon-stimulated gene 15 and its activating enzyme UBA7. *Infection, Genetics and Evolution* 56: 44-53

Minor ones:

1. The line numbers are missing in the manuscript, so it is difficult to pinpoint the questions in the text.

We added line numbers as required.

2. Some descriptions in the figure legends including the mouse numbers, infection dose etc. are not clear enough and the readers need to come back to the results/methods from time to time.

We completed the information in the figure legends.

3. In Fig. 1c, the percentage of body weight (present body weight/original body weight) is easy to read.

We updated these graphs with the % of body weight (Figure 1).

4. In Fig. 1d&e, some of the statistics are missing.

We added statistics information were applicable in this figure.

5. In Supplementary Fig. 1c and Fig. 3c, the statistics are missing.

We added statistics information were applicable in these figures.

10th Aug 2020

Dear Dr. Bourhy,

Thank you for the submission of your revised manuscript to EMBO Molecular Medicine. I am pleased to inform you that we will be able to accept your manuscript pending the following final amendments:

1) Please discuss the results observed on the viral load and the innate immune mediators by the treatment with the LALA mutant versus the non-mutated antibody as suggested by the referee #1.
2) Figures:

- Please explain the identical Kaplan Meier curves in Figure 1A, 1B and EV3E for infected non treated.

***** Reviewer's comments *****

Referee #1 (Remarks for Author):

Overall the authors have addressed the different points raised by the reviewers.

Only one point related to the LALA mutant antibodies has partially been addressed : the "evaluation the role of microglia and/or innate immune cells in Fc-mediated viral control" has not been included in the revised version of the manuscript. However, new data on brain damage and inflammation in LALA-mutant mAb-treated mice has been included. A discussion comparing the results observed on the viral load and the innate immune mediators by the treatment with the LALA mutant versus the non-mutated antibody would have been informative.

1) Please discuss the results observed on the viral load and the innate immune mediators by the treatment with the LALA mutant versus the non-mutated antibody as suggested by the referee #1.

We discussed this point as required in lines 203-205 and 213-219: "The RVC20-LALA/RVC58-LALA mAbs retained the same neutralizing activity as the non-mutated (wild-type) RVC20/RVC58 mAbs (Fig EV3D). However, the treatment with the RVC20-LALA/RVC58-LALA cocktail failed to protect mice when started at 8 dpi, and promoted only a 20% survival (1/5) when administered at 7 dpi (Fig EV3E-G, Appendix Figure S3), indicating that the Fc portion does play an important role in rabies virus clearance, as it seems to be the case in other viral infections (Chauhan et al., 2017), and in some observations of rabies virus clearance in mice (Hooper et al., 2009, Huang et al., 2014), dogs (Gnanadurai et al., 2013) and humans (de Souza & Madhusudana, 2014), where high titers of virus neutralizing antibodies were detected in the cerebrospinal fluid. Analyzing the animals that succumbed to the infection during the course of the treatment, the ones that received the non-mutated RVC20/RVC58 mAbs tended to present lower viral loads and an intriguing pattern of pro-inflammatory cytokines expression, with increased levels of IL-1 β and TNF- α , and reduced levels of IL-6 (Fig 2E-N). Conversely, in the brain of RVC20-LALA/RVC58-LALA mAbs-treated mice, viral loads and pro-inflammatory cytokines expression were similar as non-treated mice (Appendix Figure S3). In a comparative manner, in neurodegenerative diseases, the antibody-mediated clearance of *tau* protein by microglia *in vitro* is dependent on Fc-gamma receptor binding (Andersson et al., 2019); and Fc-gamma receptor is key to antibody uptake by neurons (Congdon et al., 2013). Consequently, the effective rabies virus clearance by the RVC20/RVC58 mAbs may be reinforced by Fc-gamma receptor binding, by permitting the access of the mAbs inside infected neurons, and by modulating microglial activity, inflammatory mediators production, or even by promoting the infiltration of leukocytes (Chauhan et al., 2017, Huang & Sabatini, 2020, Quan et al., 2009)."

2) Figures:

- Please explain the identical Kaplan Meier curves in Figure 1A, 1B and EV3E for infected non-treated.

The Kaplan Meier curves in Figures 2A and 2B show the same data, but presented under different aspects, 2A: according to the treatment start; 2B: according to the clinical phase. In Figure EV3E, the infected non-treated animals are the same as in Figures 2A and 2B (concomitant experiments). We added this information in the legend of Figure EV3E (lines 674-675).

The authors performed the requested changes.

Corresponding Author Name: Hervé Bourhy

Manuscript Number: EMM-2020-12628